# Term Idiopathic Polyhydramnios, and Labor Complications

**DOI:** 10.3390/jcm12030981

**Published:** 2023-01-27

**Authors:** Maayan Bas Lando, Marnina Urman, Yifat Weiss, Naama Srebnik, Sorina Grisaru-Granovsky, Rivka Farkash, Hen Y. Sela

**Affiliations:** Shaare Zedek Medical Center, Department of Obstetrics and Gynecology, Faculty of Medicine, Hebrew University of Jerusalem, Jerusalem 91031, Israel

**Keywords:** polyhydramnios, labor complications, macrosomia

## Abstract

**Background and Aim**: Polyhydramnios is associated with an increased risk of various adverse pregnancy outcomes, yet complications during labor have not been sufficiently studied. We assessed the labor and perinatal outcomes of idiopathic polyhydramnios during term labor. **Methods**: Retrospective cohort study at a tertiary medical center between 2010 and 2014. Women with idiopathic polyhydramnios defined as an amniotic fluid index (AFI) greater than 24 cm or a deep vertical pocket (DVP) > 8 cm (cases) were compared with women with a normal AFI (5–24 cm) (controls). **Statistics**: Descriptive, means ± SDs, medians + IQR. Comparisons: chi-square, Fisher’s exact test, Mann–Whitney Test, multivariate logistic models. **Results**: During the study period 11,065 women had ultrasound evaluation completed by a sonographer within two weeks of delivery. After excluding pregnancies complicated by diabetes (pre-gestational or gestational), fetal anomalies, IUFD, multifetal pregnancies, elective cesarean deliveries (CD) or missing data, we included 750 cases and 7000 controls. The degree of polyhydramnios was mild in 559 (75.0%) cases (AFI 24–30 cm or DVP 8–12 cm), moderate in 137 (18.0%) cases (30–35 cm or DVP 12–15 cm) and severe in 54 (7.0%) cases (AFI >35 cm or DVP > 15 cm). Idiopathic polyhydramnios was associated with a higher rate of CD 9.3% vs. 6.2%, *p* = 0.004; a higher rate of macrosomia 22.8% vs. 7.0%, *p* < 0.0001; and a higher rate of neonatal respiratory complications 2.0% vs. 0.8%, *p* = 0.0001. A multivariate regression analysis demonstrated an independent relation between polyhydramnios and higher rates of CD, aOR 1.62 (CI 1.20–2.19 *p* = 0.002) and composite adverse neonatal outcome aOR 1.28 (CI 1.01–1.63 *p* = 0.043). Severity of polyhydramnios was significantly associated with higher rates of macrosomia and CD (*p* for trend <0.01 in both). **Conclusions**: The term idiopathic polyhydramnios is independently associated with macrosomia, CD and neonatal complications. The severity of polyhydramnios is also associated with macrosomia and CD.

## 1. Introduction

Amniotic fluid volume assessment at term using prenatal ultrasound is common for various indications. Examples include fetal biophysical profile studies, assessment of post term pregnancies or decreased fetal movements and when assessing fetal weight. Variations in amniotic fluid volume have been considered an indicator of adverse perinatal outcome [1,2,3,4,5,6].

Polyhydramnios prevalence ranges from 0.2% to 2.0% [7]. It is defined by either an amniotic fluid index (AFI) greater than 24 cm or a deep vertical pocket (DVP) greater than 8 cm. [8] It is further subdivided to mild (AFI 24–29.9 cm or a DVP of 8–11.9 cm), moderate (AFI 30–34.9 cm or a DVP of 12–14.9 cm) and severe (AFI >35 cm or DVP >15 cm) [7,8].

Maternal, fetal and placental conditions associated with polyhydramnios include maternal diabetes mellitus, rhesus iso-immunization, congenital and chromosomal abnormalities and multiple gestation. However, in as many as 70.0% of cases, no cause is found antenatally and the polyhydramnios is referred to as idiopathic [9].

Previous reports have shown polyhydramnios to be associated with intrauterine fetal death (IUFD), preterm delivery, an unstable lie, malpresentation, cord prolapse and placental abruption [7,10,11].

However, studies examining the association between idiopathic polyhydramnios, its severity and the fetal and maternal outcome during labor are scarce. We therefore undertook this study to assess the perinatal outcome of term idiopathic polyhydramnios.

## 2. Methods

We performed a retrospective cohort study of all women with a singleton pregnancy who delivered at term (37.0–42.0 weeks of gestation). All women included in the study had an ultrasound completed in our hospital either as they were having antenatal care within our system, or had a labor and delivery triage visit due to various reasons such as false labor, preterm labor and so on. Hence, the diagnosis of polyhydramnios or the normal amount of amniotic fluid was based on an ultrasound performed at our hospital within 14 days of term delivery. We aimed to assess the perinatal outcome of term idiopathic polyhydramnios. Our clinical practice regarding management of term polyhydramnios has changed over time, specifically since the publication of the landmark study by Pilliod et al. [11] from 2015 that suggested a much higher risk of IUFD at term polyhydramnios. Up until 2015, Polyhydramnios was not a reason for induction of labor; however, since 2015 we offered and suggested induction of labor for woman with polyhydramnios based on the severity and gestational age. Since our aim in this study was to assess the natural perinatal outcome of term idiopathic polyhydramnios, we included women who delivered either spontaneously or had induction of labor for other reasons between 2010 and 2014. We excluded women who delivered pre-term (less than 37.0 weeks) or post-term (42.1 weeks and on), women with fetal anomalies (detected prenatally and postnatally), women with intra-uterine fetal death (IUFD), multifetal pregnancies, women who underwent elective cesarean deliveries (CD), women diagnosed with diabetes (pre-gestational or gestational) and women with missing data.

Data were extracted de-identified from our computerized database, which includes participants’ demographics, procedures, diagnoses and other pertinent coding, all of which was extracted from the electronic medical record and are updated during hospital stay.

Women with idiopathic polyhydramnios were defined as those with amniotic fluid index (AFI) greater than 24 cm or a deep vertical pocket (DVP) > 8 cm (cases) and were compared with women with a normal AFI (5–24 cm) (controls). Ultrasound examinations were performed by either a registered and trained sonographer or a maternal fetal medicine-trained physician, using GE-Voluson E8, Expert, 6–12 MgH or voluson 730.

### 2.1. Setting

Shaare Zedek Medical Center is a university-affiliated tertiary medical center with a large obstetric service. Roughly fifteen thousand deliveries are attended to annually. National Health and Drug Insurance plans cover all women for antenatal and peripartum care. At our medical center the real-time computerized medical records are continuously updated in real-time during labor and delivery by attending healthcare professionals. The data are audited intermittently by trained technical personnel to guarantee validity of the data. Demographic and obstetric characteristics, as well as outcome data as defined and detailed below, were extracted from the electronic database management software. Hence, for this study we did not use diagnosis codes, but rather the diagnosis as registered in the database, based on course of labor, postpartum course and the discharge letters of the mother and the newborn.

### 2.2. Definition of Measures and Outcomes 

The primary outcome was the mode of delivery: vaginal delivery (spontaneous or assisted) vs. cesarean delivery (CD). Secondary outcomes were maternal and neonatal adverse events. Maternal adverse events included postpartum hemorrhage (PPH, estimated blood loss > 500 mL and/or hemoglobin drop3 g/dL), chorioamnionitis, placental abruption, shoulder dystocia, perineal tear grade 3 and 4 (OASIS) and readmission to the hospital within 6 weeks. Definitions: Obstetrical anal sphincter injury (OASIS) classified as perineal tear grade 3 which involve the external and/or internal sphincter or perineal tear grade 4 which involve the rectal mucosa. OASIS is diagnosed by an obstetrician and corrected in the operating room. Postpartum hemorrhage (PPH) blood loss of over 500 mL after vaginal delivery and over 1000 mL after CD.

Neonatal adverse events included the following: 5-min Apgar score < 7, neonatal intensive care unit (NICU) admission for >72 h, macrosomia (birthweight > 4000 g), jaundice (defined as a need for phototherapy), Hypoglycemia (<40 mg/dL), respiratory complications (defined as transient neonatal tachypnea or respiratory distress syndrome), metabolic complications, sepsis, asphyxia, convulsions, necrotizing enterocolitis (NEC), clavicle fracture, intraventricular hemorrhage (IVH) and neonatal death. Neonatal adverse events were derived from newborn discharge records as documented by the neonatologist. Composite adverse neonatal outcome were defined as the presence of one or more of the neonatal adverse events.

### 2.3. Statistical Analyses

Descriptive continuous variables were reported as median and interquartile ranges or mean ± SD. Categorical variables were reported as counts and proportion. Univariate analyses included Chi-square test or Fisher exact test to compare between categorical variables. The effect of polyhydramnios on continuous variables was tested by unpaired student T-test or Mann–Whitney test. The choice of a parametric or nonparametric test depended on the distribution of a continuous variable. 

The “dose–response” relationship between the severity of polyhydramnios and outcomes was tested and reported with *p*-for-trend. Subgroup analysis was performed in order to assess the impact of polyhydramnios on CD in nulliparous deliveries. 

Two multivariable logistic regression analyses were developed in order to check for an independent association between polyhydramnios and (1) CD and (2) any neonatal complication. Covariates in the models included maternal age, ethnicity, hypertensive disorder of pregnancy, nulliparity, prior miscarriages, prior CD, induction of labor, use of epidural analgesia, oxytocin augmentation during labor and extreme neonatal weights defined as small for gestational age (>10th percentile) and large for gestational age (<90th percentile) based on Israeli neonatal birthweight charts by Dollberg S. et al. [12].

The discriminatory power of the models were examined using c statistics (AUC). Odds ratios (OR) with 95% confidence intervals (CIs) were reported.

All comparisons were two tailed, and *p* < 0.05 was considered statistically significant. Analyses were carried out using IBM SPSS version 22 statistical package. 

The study protocol was approved by the institutional Helsinki Committee (IRB number P24.15).

## 3. Results

During the study period, a total of 11,065 women had ultrasound evaluation completed within two weeks of delivery. There were 9919 women (89.6%) who had normal or low amniotic fluid and 1146 women (10.4%) who had polyhydramnios. After applying exclusion criteria, there were 7000 women (90.3%) with normal AFI (controls) and 750 women (9.7%) with polyhydramnios (study group). The degree of polyhydramnios was mild in 559 (75.0%) cases, moderate in 137 (18.0%) cases and severe in 54 (7.0%) cases.

There were no significant differences in the polyhydramnios rate over the study period (range of 8.5–11.2%), *p* = 0.191. 

Clinical and demographic characteristics of women are presented in Table 1. Women in the study group were older than women in the control group, with a mean age of 29.5 ± 5.7 years vs. 28.4 ± 5.7 years, (*p* < 0.0001). The percent of women at advanced maternal age (age >35 years) was also higher in the study group as compared to the control group 17.3% vs. 12.9%, (*p* = 0.001). Women in the study group were less likely to be nulliparous, with 23.0% compared to 35.0% in the control group (*p* < 0.0001). The median [IQR] time between the ultrasound and the delivery was similar in both groups.

Primary outcome: Women with polyhydramnios had a higher incidence of CD: 9.3% vs. 6.2% *p* = 0.004. Furthermore, primiparous women with polyhydramnios compared to multiparous women had a higher incidence of CD 19.1% vs. 10.1% (*p* < 0.0001). The CD rate was also higher in woman with moderate/severe polyhydramnios vs. those with a normal amount of amniotic fluid or mild polyhydramnios (12.6% vs. 6.4% *p* = 0.001). Indications for CD differed between the groups and are presented in Table 2. Arrested labor was the leading cause of CD in the study group (44.3% vs. 37.7% 0.294), whereas non-reassuring fetal heart rate was the leading cause of CD in the control group (47.8% vs. 30.0% *p* = 0.005). There were two cases of umbilical cord prolapse in women with polyhydramnios (0.3%) and five cases in the control group (0.07%). Severity of polyhydramnios was significantly associated with higher rates of macrosomia and CD (p for trend <0.01 in both) as presented in Figure 1. Multivariate regression model accounting for various characteristics as detailed in the method section revealed the following: polyhydramnios was independently associated with higher rates of CD, at an adjusted odds ratio (aOR) of 1.62 (CI 1.2–2.2 *p* = 0.002) AUC 0.83 (95%CI 0.81–0.85).

Labor characteristics are presented in Table 3. The median gestational age at birth was 40 (39–41) weeks in both groups (*p* = 0.546). The median duration of the first stage of labor was similar between the groups (239 min vs. 219 min *p* = 0.101), whereas median duration of the second stage of labor was shorter in the study group (110 vs. 99 min *p* = 0.005). The rate of meconium-stained amniotic fluid was higher in study group 21.1% vs. control group 17.7% *p* = 0.023. There were no differences between the groups regarding the rates of induction or augmentation of labor, use of epidural analgesia or rate of chorioamnionitis, placental abruption, shoulder dystocia, perineal tear grades 3 and 4 or post-partum hemorrhage. A composite adverse maternal outcome (including CD, PPH, placental abruption, perineal tear grades 3 and 4, hemoglobin drops more than 3 g, shoulder dystocia, chorioamnionitis and readmission to the hospital within 6 weeks) was higher in study group vs. control group (164 (21.9%) vs. 1266 (18.1%) *p* = 0.011).

Neonatal characteristics and outcomes are presented in Table 4. Mean birthweight of newborns was significantly higher in the study group (3640 ± 467 gr vs. 3334 ± 470 gr (*p* < 0.0001)), and there was a higher rate of macrosomia in the study group (22.8% vs. 7.0%, (*p* < 0.0001)). The macrosomia rate was also significantly more common in women with moderate/severe polyhydramnios vs. those with a normal amount of amniotic fluid or with mild polyhydramnios (31.4% vs. 8% *p* < 0.0001). There was a higher rate of neonatal respiratory complications in the study group 2.0% vs. 0.8%, (*p* < 0.0001). There were no differences between the groups in the rate of NICU admission > 72 h 1.1% vs. 0.7% (*p* = 0.25) and low Apgar scores (defined as Apgar score <7 at 5 min) 0.9% vs. 0.6% (*p* = 0.202).

A composite adverse neonatal outcome occurred in 12.5% of mild polyhydramnios, 13.9% of moderate polyhydramnios and 7.4% of severe polyhydramnios compared to 9.3% for controls. There were no neonatal deaths in either group. Multivariate regression models accounting for various characteristics as detailed in the method section revealed that polyhydramnios was independently associated with composite adverse neonatal outcome, aOR of 1.28 (CI 1.01–1.63 *p* = 0.043) AUC 0.58 (95%CI 0.56–0.6).

## 4. Discussion

In the present study, we evaluated the labor, maternal and neonatal outcomes of term idiopathic polyhydramnios. Our study found that term idiopathic polyhydramnios was associated with a higher rate of CD, macrosomia and neonatal respiratory complications. These findings remained significant after controlling for confounders by using multivariable logistic regression analysis, which revealed that idiopathic polyhydramnios was associated with an increased risk of CD and an increased risk of composite adverse neonatal outcomes. 

Previous reports of isolated polyhydramnios and labor outcomes are conflicting. The differences may be explained by the heterogenic group of patients and various designs and inclusion criteria. For example, some studies included cases with diabetes and fetal anomalies or evaluated term and preterm deliveries together. 

In our study, polyhydramnios was associated with macrosomia, similar to the findings by Dorleijn et al. [9] as well as Yefet E and Daniel-Spiegel E [13]. In the latter study, polyhydramnios with a normal detailed anatomy scan was associated with increased risk for CD and a birth weight in the >90th percentile. This increase in CD was attributed to the higher rate of elective CD due to suspected macrosomia. Our study also demonstrated an increase in CD rate due to an increase in macrosomia and arrested labor. Similarly, Magann et al. [5] showed that CD for fetal distress were more common with polyhydramnios as well as low 5-min Apgar scores, increased neonatal birthweight and newborn intensive-care unit admissions. Morris et al. [14] also showed in a meta-analysis that polyhydramnios is strongly associated with a birthweight in the >90th percentile. Interestingly, Asadi et al. [15] found an increase in low birth weight (<2500 g) as well as macrosomia (>4000 g) in their case group, perhaps due to a difference in selection of the study population. 

In our study, the duration of the first stage of labor was similar between the groups, whereas median duration of the second stage of labor was shorter in the study group. There were no differences between the groups regarding the rates of induction or augmentation of labor, use of epidural analgesia or rate of chorioamnionitis, placental abruption, shoulder dystocia, perineal tear grades 3 and 4 or post-partum hemorrhage.

Aviram et al. [16] showed that polyhydramnios was an independent risk factor for labor induction, CD, prolonged first stage of delivery, abnormal or intermediate FHR tracings, placental abruption, shoulder dystocia and respiratory distress. Mild isolated polyhydramnios was independently associated with CD, prolonged first stage of delivery, placental abruption, abnormal or intermediate FHR tracings and shoulder dystocia. Khan and Donnelly [17] found a higher rate of caesarean delivery, fetal distress and NICU admissions.

Zeino et al. [18] did not observe differences in the rate of epidural analgesia or rate of abnormal fetal heart tracing. Induced labor, amniotomy and non-vertex presentations were more frequent in their polyhydramnios group. In their study, CD rate was higher in pregnancies with polyhydramnios and remained higher after exclusion of induced labor and non-vertex presentation.

Another important finding in our present study is the increase in perinatal respiratory morbidity, defined as transient neonatal tachypnea or respiratory distress syndrome without a significant difference in other morbidities examined. However, as there was no difference in NICU admission rate for more than 72 h, it is possible that these were non-severe respiratory complications. We also demonstrated no increase in fetal or perinatal death. Chauhan et al. [19] showed that polyhydramnios was significantly associated with the delivery of a macrosomic fetus, but not with delivery of a compromised neonate or CD. Thompson et al. [20] showed that there was a better overall fetal outcome compared with previous studies and no perinatal deaths. Pasquini et al. [21] also did not find significant difference in the Apgar score and the rate of neonatal hypoxia in mild idiopathic polyhydramnios.

Conversely, Maymon et al. [22] showed that the presence of polyhydramnios was strongly associated with perinatal mortality and neonatal and maternal morbidity. Similarly, Asadi et al. [15] found that NICU admission, fetal distress, fetal death, lower 1-min and 5-min APGAR scores, preterm delivery and neonatal death were higher in their polyhydramnios cohort as compared to controls. Chen et al. [4] showed that polyhydramnios carried a higher incidence of adverse perinatal outcomes, such as low Apgar scores, fetal death, fetal distress in labor, NICU transfer and neonatal death, despite the exclusion of congenital anomalies from the study. Likewise, Polnaszek et al. [23] showed that polyhydramnios increased the risk of composite neonatal morbidity, including respiratory morbidity. Additionally, Pagan et al. [24] showed that idiopathic polyhydramnios had higher odds of neonatal death, intrauterine fetal demise, NICU admission and a 5 min APGAR score less than 7. 

Importantly, our study demonstrates that the severity of polyhydramnios is independently associated with a higher risk of macrosomia and CD. Our results are similar to those of Harlev et al. [25], who showed a significant linear association between the AFI and adverse pregnancy outcome, including hypertensive disorders, diabetes mellitus, preterm labor, macrosomia, placental abruption and low birth weight. A different population selection may cause the difference. 

A possible mechanism underlying this finding may be overdistention of the uterus caused by polyhydramnios, making it less responsive to oxytocin as the polyhydramnios increases in severity, leading to hypotonic disorders of labor progression and a need for CD. Conversely, an association between severity and outcome was not demonstrated in a study by Asadi et al. [15], who found no significant statistical differences in fetal and maternal adverse outcomes according to the severity of polyhydramnios.

Our analysis of demographic data showed that polyhydramnios was more common in older women and multipara similar to the data of Biggio et al. [26], who found a relationship between idiopathic polyhydramnios and increasing maternal age and parity.

Although premature rupture of membranes (PROM) has been related to polyhydramnios, in our study, PROM was less common in the study group of idiopathic polyhydramnios than the control group. A possible mechanism underlying this finding may be the high incidence of PROM of around 10% at term with the general population of pregnant woman [27].

Indeed, the rate of polyhydramnios in our study is much higher than what has been previously reported, yet it should be noted that the true prevalence is unknown.

Our study has strengths and limitations. The limitations include the following: this was a retrospective study and has the inherited limitations of retrospectively collected information. During the study period, there were roughly 50,000 term singleton deliveries; however, only 20% of those had an ultrasound exam in our institution within 2 weeks of delivery. It is, therefore, possible that our study suffers from selection bias. As noted, there are several differences between study groups; for instance, 23% of women with polyhydramnios were nullipara, vs. 35% of the women in the control group. We do not have data regarding the BMI of parturients as this information was not routinely collected during these years. We do not have information regarding infection screening as a cause for polyhydramnios, but the significance of investigating infectious causes has not been shown to be beneficial in the workup of polyhydramnios [28,29]. Furthermore, as the association between polyhydramnios and IUFD had been previously studied, we did not look at this association in this study.

The strengths of our study include the large sample size and the fact that this is a historical cohort before our departmental protocol was changed to offer induction of labor for women with idiopathic polyhydramnios at term. Even though women had an ultrasound exam within 2 weeks of delivery, the median time from ultrasound to delivery was not different between the groups. Furthermore, we included in this study only women without diabetes and without fetal/neonatal anomalies.

## 5. Conclusions

We found that term idiopathic polyhydramnios is independently associated with macrosomia CD and neonatal complications. Additionally, the severity of polyhydramnios was found to be associated with macrosomia and CD. The information from this study will assist health care providers in counseling their patients with term idiopathic polyhydramnios regarding the expected mode of delivery according to the severity of polyhydramnios, and will allow the NICU team to be prepared for possible respiratory difficulties in the neonates of this group.

## Figures and Tables

**Figure 1 jcm-12-00981-f001:**
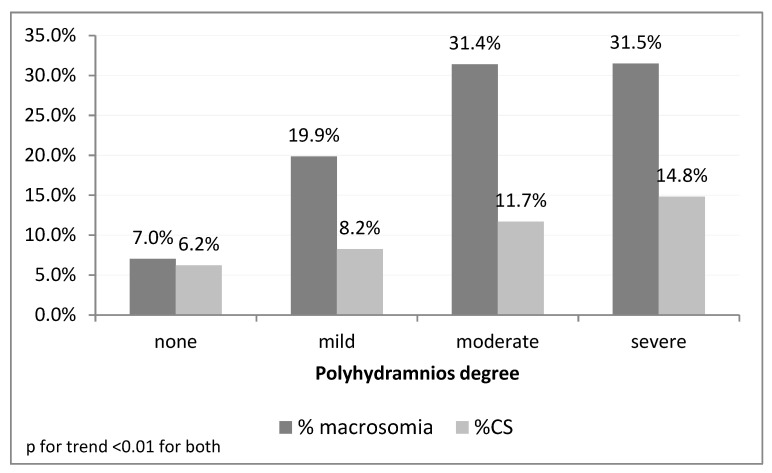
Rates of macrosomia and Cesarean delivery by polyhydramnios degree. Mild: AFI 24–29.9 cm or DVP of 8–11.9 cm; Moderate: AFI 30–34.9 cm or DVP of 12–14.9 cm; Severe: AFI > 35 cm or DVP > 15 cm.

**Table 1 jcm-12-00981-t001:** Maternal demographic and clinical Characteristics.

Characteristics	Study (*n* = 750)	Controls (*n* = 7000)	*p* Value
Maternal age (mean, years)	29.5 ± 5.7	28.4 ± 5.7	*p* < 0.0001
Advanced maternal age (>35 years, N, %)	130 (17.3)	904 (12.9)	0.001
Nulliparity (N, %)	173 (23.1)	2469 (35.3)	<0.0001
Prior cesarean delivery (CD) (N, %)	86 (11.5)	661 (9.4)	0.074
3 or more prior miscarriages (N, %)	36 (4.8)	229 (3.3)	0.029
Assisted Reproductive Technology (ART) (N, %)	28 (3.7)	245 (3.5)	0.742
Hypertensive disorder (N, %)	6 (0.8)	82 (1.2)	0.362
Time from US to delivery (median, days)	2 (1–5)	2 (1–5)	0.331

**Table 2 jcm-12-00981-t002:** Indications for cesarean deliveries.

Indications for Cesarean Delivery (N, %) Total CD = 505	Study (*n* = 70)	Controls (*n* = 435)	*p* Value
Non- reassuring fetal heart rate	21 (30)	208 (47.8)	0.005
Arrested labor (including failed induction)	31 (44.3)	164 (37.7)	0.294
Suspected Macrosomia	10 (14.3)	9 (2.1)	<0.0001
Umbilical cord Prolapse	2 (2.9)	5 (1.1)	0.26
Other (including maternal and/or fetal complications)	6 (8.5)	49 (11.3)	0.502

**Table 3 jcm-12-00981-t003:** Labor Characteristics.

Characteristics	Study (*n* = 750)	Controls (*n* = 7000)	*p* Value
Gestational age at delivery (median [IQR])	40 (39–41)	40 (39–41)	0.546
Premature rupture of membranes (N, %)	60 (8%)	818 (11.7%)	0.002
Induction of labor (N, %)	132 (17.6)	1292 (18.5)	0.565
Oxytocin during labor (N, %)	290 (38.7)	2470 (35.3)	0.066
Epidural anesthesia (N, %)	522 (69.6)	4888 (69.8)	0.897
Meconium stained amniotic fluid (N, %)	158 (21.1)	1240 (17.7)	0.023
Chorioamnionitis (N, %)	24 (3.2)	172 (2.5)	0.218
Placental abruption (N%)	10 (1.3)	64 (0.9)	0.262
Mode of delivery (N, %)			
• Spontaneous vaginal delivery	622 (82.9)	5986 (85.5)	0.004
• Instrumental vaginal delivery	58 (7.7)	579 (8.3)	
• Cesarean delivery	70 (9.3)	435 (6.2)	
Shoulder dystocia (N, %)	7(0.9)	52 (0.7)	0.568
Perineal tear grade 3,4 (N%)	6 (0.8)	42 (0.6)	0.507
Post partum hemorrhage (N%)	78 (10.4)	701 (10.0)	0.738
Composite adverse maternal outcome (N%)	164 (21.9%)	1266 (18.1%)	0.011

**Table 4 jcm-12-00981-t004:** Neonatal Characteristics and outcomes.

Characteristics	Study (*n* = 750)	Controls (*n* = 7000)	*p* Value
Birth weight (mean, grams, SD)	3640 ± 467	3334 ± 470	<0.0001
Macrosomia > 4000 g (N%)	171 (22.8)	491 (7.0)	<0.0001
Female gender (N%)	336 (44.8)	3414 (48.8)	0.039
Apgar score at 5 min < 7 (N %)	7 (0.9)	39 (0.6)	0.202
NICU admission > 72 Hours (N%)	8 (1.1)	49 (0.7)	0.25
Small for gestational age * (N %)	19 (2.5)	691 (9.9)	<0.0001
Neonatal jaundice (N %)	42 (5.6)	339 (4.8)	0.362
Hypoglycemia (N %)	31 (4.1)	207 (3.0)	0.076
Respiratory complications (N %)	15 (2.0)	55 (0.8)	0.001
Composite adverse neonatal outcome (N%)	93 (12.4%)	649 (9.3%)	<0.006

Small for gestational age * <10 percentile.

## Data Availability

Data available on request from the corresponding author due to privacy and ethical restrictions.

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
