# Peer review of "Term Idiopathic Polyhydramnios, and Labor Complications"

_jcm, 2023, doi:10.3390/jcm12030981_

Round 1
Reviewer 1 Report
ABSTRACT
Line 25: adjusted ???
METHODS
Line 55: I suppose all women visited your antenatal clinic for routine prenatal care. You may add this to the text.
Please rephrase sentence 57-62.
Line 60: At which gestational age do you offer now induction in women with hydramnios – already from 37 weeks?
Line 62: You excluded induction for hydramnios – because this was not standard management during the study period, in my opinion you should not exclude this.
Did you encounter any cases of fetal death at the last US. If so – were these excluded? Probably you better keep them in and divide between hydramnios and controls.
Line 82-90: Outcome criteria diagnosis need references or threshold values.
Line 105 : extreme neonatal weights – please give threshold.
RESULTS:
Line 123: Median gestational age – add “at ultrasound assessment” . You may also decide to omit parameters that are similar in both groups from the text, write out only the parameters with differences and then that all remaining were similar , and refer to the table (same as you did for table 2)
Table 1: Sort row captions sequentially,
Table 2: I miss gestational age at delivery, pre-labour rupture of membranes and chorioamnionitis in this table. Please left-align the row captions. Row captions can be better ordered in chronological sequence, so first induction, then delivery and end with postpartum. You may add as sub-rows under CD the CD rates for nulliparity / multiparity. How many CS were pre-labour? You may add a row with total maternal adverse outcome.
Line 132: You may add duration of 2nd stage and 3d stage to table 2
Table 3: you calculate a summarized p by multirow Chi-square. You might instead calculate p values for the separate diagnoses – I suppose prolapsed cord and “others” are not stat sig.
Table 4: Improve the row-caption order, e.g. first weight and weight classes, then sex, condition, (pH??), NICU and diagnosis. Add “composite diagnosis”. In the methods you name this adverse events. Please select a uniform caption for this, e.g. composite adverse infant outcome” – same for maternal outcome.
Table 4: Respiratory complications were twice as frequent as NICU admission in the hydramnios group, but similar in the controls. Apparently there was a difference in severity, with more minor cases in the study group. As requested in methods, the diagnosis in the table should have a definition and classification. Were all infants standard admitted to medium/low risk neonatal care after delivery or could you differentiate between these modalities? Most cases of jaundice and hypoglycemia did not need NICU, so here the same question regarding treatment. Generally jaundice and hypoglycemia are only relevant if above a certain threshold, which depends on age, and need treatment.
Line 162: hydramnios irrespective of grade ? You did not assess normal/mild vs moderate/severe?
Line 162-163: You might present AUC (95% interval) as an sign of efficacy of the model.
DISCUSSION
Line 182: Please add citation number directly after the authors Yefet E and Daniel-Spiegel E.
Line 186: Table 3 shows the opposite??? Furthermore, you compare elective CS with unspecified CS.
Line 188: Add citation number after author name, same for later references.
192: Difference in selection of the study population by ref 24?
Line 211: this needs some classification/definition.
Line 230: Here a different population selection may cause the difference of rf 19. Please specify this.
Line 234: You may be right here, but then I would expect also more PPH
Line 235: “rapid reduction of size” – I would expect this only to occur at rupture of membranes. You may remove this sentence on abruptio – your study did not show an increase of abruptio and those who did might have a different selection (possibly different GA?)
Line 240: add “and multipara”
Line 243: PROM is not “premature rupture of membranes” but “pre-labour rupture of membranes”. If it is premature it is called PPROM.
Line 245: You do not show PROM in your data.
These data do not support your later change of management to offer induction to women with hydramnios at term, or were term fetal death excluded? You may comment on this in the discussion.
Author Response
Thank you for the opportunity to resubmit our revised manuscript entitled: “Term Idiopathic Polyhydramnios, and labor complications" for publication in Journal of Clinical Medicine.
We appreciate the opportunity to address the reviewers' comments and questions and submit our revised version. We are certain that incorporating these changes has improved the manuscript.
Attached are our point-by-point responses to the comments raised. In our revised manuscript, we highlighted the corresponding changes with track changes.
- ABSTRACT Line 25: adjusted ???
Indeed, these were adjusted OR – we have now corrected it in lines 25 and 26 to be aOR.
- METHODS Line 55: I suppose all women visited your antenatal clinic for routine prenatal care. You may add this to the text. Indeed, all women had an ultrasound done in our hospital with in 14 days of term delivery either as they were having antenatal care within our system or had a labor and delivery triage visit due to various reasons such as false labor, preterm labor and so on. Hence the diagnosis of polyhydramnios or normal amount of amniotic fluid was based on ultrasound performed at our hospital within 14 days of term delivery.
- Please rephrase sentence 57-62.
We appreciate this comment and have rephrased this paragraph – hopefully it would be more self-explanatory now: Our clinical practice regarding management of term polyhydramnios has changed over time, specifically since the publication of landmark study by Pilliod et al [11] from 2015 that suggested a much higher risk of IUFD at term polyhydramnios. Up until 2015 Polyhydramnios was not a reason for induction of labor, however since 2015 we offered and suggested induction of labor for woman with polyhydramnios based on the severity and gestational age. Since our aim in this study was to assess the natural perinatal outcome of term idiopathic polyhydramnios, we included women who delivered either spontaneously or had induction of labor for other reasons between 2010 and 2014.
- Line 60: At which gestational age do you offer now induction in women with hydramnios – already from 37 weeks? Following the study by Pilliod et al that included nearly 2 million births in the USA with nearly 7000 pregnancies complicated by polyhydramnios with a 7 folds increased risk at 37 weeks and 11 folds increased risk by 40 weeks we now offer induction of labor for severe polyhydramnios at gestational age 37-39 and for mild and moderate polyhydramnios at gestational age 39-40.
- Line 62: You excluded induction for hydramnios – because this was not standard management during the study period, in my opinion you should not exclude this. Thank you for this comment. The exclusion criteria is detailed in the text. During the years of the study we did not induce women due to polyhydramnios. Indeed, we could have performed a study that compared pre and post clinical management change (IOL for Poly), however as not all woman agree to be induced per our advice we felt that the post change period would not be ‘clean’ as such and according to aim of this study the assess the natural perinatal outcomes of women with term polyhydramnios we proceeded with the study as detailed.
- Did you encounter any cases of fetal death at the last US. If so – were these excluded? Probably you better keep them in and divide between hydramnios and controls. We thank you for the opportunity to clarify this point – the association between polyhydramnios and IUFD had been studied. In our clinical practice setting it is possible that woman presented with IUFD, however did not have an ultrasound assessment within 14 days of the IUFD – hence we could not categorized her to normal amniotic fluid or polyhydramnios hence we excluded woman with IUFD. This is now added to the exclusion criteria and limitation section
- Line 82-90: Outcome criteria diagnosis need references or threshold values. Thank you for this comment and again the opportunity to clarify. Data on demographic and obstetric characteristics, as well as outcome data as defined and detailed bellow were extracted from the electronic database management software. At our medical center the real-time computerized medical records are continuously updated in real-time during labor and delivery by attending healthcare professionals. The data is audited intermittently by trained technical personnel to guarantee validity of the data. Hence for this study we did not use diagnosis codes rather diagnosis as registered in the database, based on course of labor, postpartum course, discharge letters of the mother and the newborn. We now added a section the setting subsection detailing that and further detailed regarding the outcomes diagnosis.
- Line 105: extreme neonatal weights – please give threshold. We added to the manuscript: small for gestational age >10th percentile and large for gestational age <90th percentile based on Israeli neonatal birthweight charts (Dollberg S et. al. 2005).
RESULTS:
- Line 123: Median gestational age – add “at ultrasound assessment” . You may also decide to omit parameters that are similar in both groups from the text, write out only the parameters with differences and then that all remaining were similar , and refer to the table (same as you did for table 2). We, as suggested, added to the manuscript median gestational age at delivery and moved it to table 2 and we also omitted parameters that are similar in both groups from the text. We did not use median gestational age at the time of ultrasound assessment, yet we have calculated time interval between ultrasound assessment and delivery and this is now presented in the table.
- Table 1: Sort row captions sequentially. Thanks for the comment, please see revised in table 1.
- Table 2: I miss gestational age at delivery, pre-labour rupture of membranes and chorioamnionitis in this table. Please left-align the row captions. Row captions can be better ordered in chronological sequence, so first induction, then delivery and end with postpartum. You may add as sub-rows under CD the CD rates for nulliparity / multiparity. How many CS were pre-labour? We appreciate all these comments and have now added to the manuscript median gestational age at delivery in table 2, PROM and chorioamnionitis to the table. we left-aligned the row captions and placed them in chronological sequence We excluded elective CS. We added a row with composite adverse maternal outcome in table 2.
- Line 132: You may add duration of 2nd stage and 3d stage to table 2. Thank you for this comment. Duration of 1st and 2nd stage of labor are written in the text and table 2 might be too long if will add them to the table. We do not have data about the 3rd stage of labor.
- Table 3: you calculate a summarized p by multirow Chi-square. You might instead calculate p values for the separate diagnoses – I suppose prolapsed cord and “others” are not stat sig. Thank you for this comment. We calculated p values for the separate diagnoses and have revised the table accordingly. this table is mainly descriptive as the rate of total CD was low.
- Table 4: Improve the row-caption order, e.g. first weight and weight classes, then sex, condition, (pH??), NICU and diagnosis. Add “composite diagnosis”. In the methods you name this adverse events. Please select a uniform caption for this, e.g. composite adverse infant outcome” – same for maternal outcome. We appreciate this comment and have revised the table accordingly, please see revised Table 4.
- Table 4: Respiratory complications were twice as frequent as NICU admission in the hydramnios group, but similar in the controls. Apparently there was a difference in severity, with more minor cases in the study group. As requested in methods, the diagnosis in the table should have a definition and classification. Were all infants standard admitted to medium/low risk neonatal care after delivery or could you differentiate between these modalities? Most cases of jaundice and hypoglycemia did not need NICU, so here the same question regarding treatment. Generally jaundice and hypoglycemia are only relevant if above a certain threshold, which depends on age, and need treatment. Thank you for this comment. Indeed, NICU admissions for more than 72 hours was not statically different between groups (1.1% vs 0.7%, P=0.250) while respiratory complications were statically more common in the study group (2% vs 0.8%, p=0.001) indeed may indicate mostly mild cases of respiratory complications – this is now added to the discussion section. NICU admission and length of stay at NICU is derived from the hospital administrative data, while neonatal adverse events were derived from newborn discharge records as documented by the neonatologist. Even though hypoglycemia and Jaundice are indeed clinically meaningful only at certain degree, nevertheless they are diagnosis that are registered at the newborn discharge letter hence we included as outcomes.
- Line 162: hydramnios irrespective of grade? You did not assess normal/mild vs moderate/severe? Thank you for this question, indeed, CD rate were higher in woman with moderate/severe polyhydramnios vs those with normal/mild polyhydramnios (12.6% vs 6.4% p=0.001) we now added this to the text.
- Line 162-163: You might present AUC (95% interval) as an sign of efficacy of the model. Much appreciate, we added multivariate logistic regression analysis with CD as outcome: AUC 0.83 (95%CI 0.81-0.85) and Multivariate logistic regression analysis with any neonatal complication as outcome: AUC 0.58 (95%CI 0.56-0.6).
DISCUSSION
- Line 182: Please add citation number directly after the authors Yefet E and Daniel-Spiegel E. Done.
- Line 186: Table 3 shows the opposite??? Furthermore, you compare elective CS with unspecified CS. Thank you for this comment. We corrected in the manuscript: Our study also demonstrated an increase in CD rate due to an increase in macrosomia and arrested labor.
- Line 188: Add citation number after author name, same for later references.
- Line 192: Difference in selection of the study population by ref 24? Thank you for your hypothesis. we agree, and we added that perhaps due to a difference in selection of the study population
- Line 211: this needs some classification/definition. We added Respiratory complications defined as transient neonatal tachypnea or respiratory distress syndrome and we have further added that as NICU admission rate for more than 72 hours were not different, it is likely minor respiratory disease.
- Line 230: Here a different population selection may cause the difference of rf 19. Please specify this. Indeed, we specified: a different population selection may cause the difference.
- Line 234: You may be right here, but then I would expect also more PPH. Thank you for the comment. It is very interesting that the rate of PPH was similar, we may speculate that this was due to team anticipation of PPH and using preventive meaures, as active management of third stage of labor.
- Line 235: “rapid reduction of size” – I would expect this only to occur at rupture of membranes. You may remove this sentence on abruptio – your study did not show an increase of abruptio and those who did might have a different selection (possibly different GA?). We removed this sentence on abruptio as offered.
- Line 240: add “and multipara”. Done.
- Line 243: PROM is not “premature rupture of membranes” but “pre-labour rupture of membranes”. If it is premature it is called PPROM. Unfortunately on this point we disagree, please see Gabbe's Obstetrics: Normal and Problem Pregnancies 8th Edition 2020, chapter 37 Premature Rupture of Membranes by Brian M Mercer – abbreviation table on page 694 s well as first paragraph of the chapter -PROM “Premature Rupture Of Mebranes” – please see Gabbe.
- Line 245: You do not show PROM in your data. Indeed, we now added PROM to table 2.
- These data do not support your later change of management to offer induction to women with hydramnios at term, or were term fetal death excluded? You may comment on this in the discussion. Please see our response to comment number 6 – we have now added to the discussion an explicit sentence that this study was not aimed to assess risk of IUFD.
Reviewer 2 Report
1. Your references are dated as most recent is 2018 is this the best you can do?
2. Your definition of idiopathic line 71-75 is incomplete as you have not included all your exclusions such diabetes etc.
3. Table 1 no explanation for nulliparity differences and how is the miscarriage difference important.
4. Generally good but Discussion could be improved for the readers benefit as a bit wandering.
Author Response
Thank you for the opportunity to resubmit our revised manuscript entitled: “Term Idiopathic Polyhydramnios, and labor complications" for publication in Journal of Clinical Medicine.
We appreciate the opportunity to address the reviewers' comments and questions and submit our revised version. We are certain that incorporating these changes has improved the manuscript.
Attached are our point-by-point responses to the comments raised. In our revised manuscript, we highlighted the corresponding changes with track changes.
- Your references are dated as most recent is 2018 is this the best you can do? We repeated a pubmed search using the terms idiopathic polyhydramnios between 2019 and today and yielded only 27 manuscripts in English. We added several contributing maniscripts to our paper: references 21,23,24.
- Your definition of idiopathic line 71-75 is incomplete as you have not included all your exclusions such diabetes etc. Lines 62-66 in the original submitted manuscript include exclusion criteria and these include women with DM or GDM, it is currently listed in the revised submitted manuscripts in line 79.
- Table 1 no explanation for nulliparity differences and how is the miscarriage difference important. Indeed, there are some differences between women with polyhydramnios and women without polyhydramnios as noted, however we could not find a plausible explanation in the literature or a possible mechanism that might explain this. We now added this difference in population characteristics in study limitation section.
- Generally good but Discussion could be improved for the readers benefit as a bit wandering. Thank you for your comment. We have now revised the discussion based on your comments as well as Reviewer 1 and believe it is much improved.